# Impact of Voxel Normalization on a Machine Learning-Based Method: A Study on Pulmonary Nodule Malignancy Diagnosis Using Low-Dose Computed Tomography (LDCT)

**DOI:** 10.3390/diagnostics13243690

**Published:** 2023-12-18

**Authors:** Chia-Chi Hsiao, Chen-Hao Peng, Fu-Zong Wu, Da-Chuan Cheng

**Affiliations:** 1Department of Radiology, Kaohsiung Veterans General Hospital, Kaohsiung 813414, Taiwan; vghkscchsiao1@gmail.com; 2Department of Biomedical Imaging and Radiological Science, China Medical University, Taichung 40400, Taiwan; 112202701@365.cmu.edu.tw

**Keywords:** lung cancer, isotropic voxel reconstruction, lung nodule classification, radiomics, computer-aided diagnosis

## Abstract

Lung cancer (LC) stands as the foremost cause of cancer-related fatality rates worldwide. Early diagnosis significantly enhances patient survival rate. Nowadays, low-dose computed tomography (LDCT) is widely employed on the chest as a tool for large-scale lung cancer screening. Nonetheless, a large amount of chest radiographs creates an onerous burden for radiologists. Some computer-aided diagnostic (CAD) tools can provide insight to the use of medical images for diagnosis and can augment diagnostic speed. However, due to the variation in the parameter settings across different patients, substantial discrepancies in image voxels persist. We found that different voxel sizes can create a compromise between model generalization and diagnostic efficacy. This study investigates the performance disparities of diagnostic models trained on original images and LDCT images reconstructed to different voxel sizes while making isotropic. We examined the ability of our method to differentiate between benign and malignant nodules. Using 11 features, a support vector machine (SVM) was trained on LDCT images using an isotropic voxel with a side length of 1.5 mm for 225 patients in-house. The result yields a favorable model performance with an accuracy of 0.9596 and an area under the receiver operating characteristic curve (ROC/AUC) of 0.9855. In addition, to furnish CAD tools for clinical application, future research including LDCT images from multi-centers is encouraged.

## 1. Introduction

According to the 2023 World Cancer Statistics [1], although the overall mortality rate of lung cancer (LC) has experienced a decline in 2023 compared to previous years, it remains the second most prevalent cancer globally in terms of both overall cancer-related deaths and new cases. Approximately 230,000 individuals are newly diagnosed with LC each year globally, and among them about 120,000 suffer malignancy. Clinically, LC can be broadly classified into non-small-cell lung cancer (NSCLC) and small-cell lung cancer (SCLC). NSCLC is the most prevalent, accounting for approximately 85% of LC cases [2], and can further be categorized into adenocarcinoma and squamous-cell carcinoma subtypes [3]. The early detection and treatment of NSCLC typically leads to better prognoses and increases survival rates by five years P [3].

Currently, low-dose computed tomography (LDCT) is commonly used in early LC screening, enabling rapid and large-scale diagnoses for individuals with high risk factors [4]. However, these screenings generate a large number of images that require interpretation by radiologists. This not only results in a substantial burden on human resources but also introduces variations in experience and subjective awareness among different radiologists, leading to a lack of unified diagnostic criteria [5]. Within LDCT images, pulmonary nodules (PNs) represent areas of suspected lesions and can be preliminarily categorized into three types: solid nodules (SN), part-solid nodules (PSN), and ground glass nodules (GGN) [6]. Typically, radiologists rely on visible characteristics such as the PN’s size, edge shape, and nodule pattern for diagnosis [7], similar to Lung-RADS™ [8]. However, this approach often yields a high rate of false positives [9], which ultimately creates the need for a pathological biopsy for a final confirmation. This process often poses unnecessary and invasive risks to patients [10].

The use of machine learning (ML) and deep learning (DL) for pulmonary nodule detection and diagnosis from chest CT images has been studied for decades. In particular, radiomics [11] is a feature-extraction technique, that is usually combined with classifiers to establish models, which can provide physicians with an alternative opinion beyond visible characteristics. A previous study indicates that such models significantly enhance reading speed and diagnostic precision for physicians [12]. Numerous studies that focused on PN detection and classification found that these models can achieve promising results [13,14,15,16,17,18,19,20,21,22,23]. For example, Kumar et al. [16] proposed an approach utilizing autoencoders to extract features from raw images and employ a decision tree for binary classification, which achieved an accuracy of 0.7501. Shen et al. [22] introduced a method using multi-scale convolutional neural networks (MCNN) to distinguish a PN’s malignancy, capturing differently sized patches of each PN and simultaneously using convolutional neural networks (CNN) for discrimination, which achieved an accuracy of 0.8612 on the LIDC-IDRI dataset [5]. Mehta et al. [19] (2021) presented a model that utilizes 3D CNNs in combination with volumetric radiomics and imaging biomarkers to diagnose a PN’s malignancy. By incorporating 3D PN images, radiomics, and LIDC-IDRI biomarkers, they achieved an ROC/AUC of 0.8659. In 2021, Lu et al. [18] introduced a method that integrated the Marine Predators Algorithm with CNN for the diagnosis of PNs. This approach was impressive, achieving an accuracy of 0.934 and a sensitivity of 0.984. Halder et al. [24] attained the best PN diagnostic model performance using the LIDC-IDRI dataset in 2021. They employed morphology and two CNN networks and achieved an accuracy of 0.9610 and an AUC of 0.9936.

However, the above-mentioned studies face some common challenges. First, most patients in the LIDC-IDRI dataset lack pathological biopsy results on nodule malignancy diagnosis, which are considered to be the gold standard. The LIDC-IDRI data provides suspicion scores of nodule malignancy, ranging from 1 to 5, which were made by radiologists. Most previous studies classify nodules with scores of 1 and 2 as benign and those with scores of 4 and 5 as malignant, however, the actual malignancy of PNs remains uncertain. Second, previous studies typically used raw LDCT images as inputs to their models. They ignored the fact that images undergoing LDCT imaging may have had different modality settings such as pixel spacing (PS) and slice thickness (SK). This ignorance raises the question of whether the uniformity of a voxel size would affect model diagnostic performance or not. For instance, Kim et al. [25] (2019) reviewed factors that could impact the quantification of CT image features, highlighting the significant influence of PS and SK on feature quantification. Most features demonstrate different impacts before and after normalization of PS and SK. Lu et al. [26] (2017) improved model diagnostic performance by reconstructing positron emission tomography (PET) images into uniform voxel sizes. Their study offers valuable insights by investigating the impacts of different voxel sizes on PET image quantification and diagnosis. The above-mentioned studies focused on a traditional machine learning method, therefore they could examine feature impact accordingly. Recent studies often used neural networks (NN) and deep learning [27], which is outside of this study’s scope. This study design was motivated by the above-mentioned problems; therefore, we did not consider using NN and DL. Furthermore, DL models demand large datasets for training, which is not suitable the small dataset used in this study.

## 2. Materials and Methods

### 2.1. Dataset in House and Annotation

LDCT image data are collected from Kaohsiung Veterans General Hospital (KVGH), including 160 malignant and 81 benign PNs from a total of 241 patients. The data collection process has been approved by the Institutional Review Board (IRB) of KVGH with the IRB number VGHKS18-CT5-09. The malignancy status of each PN is defined by pathological biopsy and the boundary of the PN is annotated by an experienced radiological technician. LifeX (Version 6.2.0, C. Nioche, Inserm, Paris, France) is open-source software that was first published in 2018 [28]. Its application involves reading DICOM images and synchronizing the display and annotation of medical images in the coronal, sagittal, and axial slices. It also concurrently supports users in the extraction of radiomics features within regions of interest (ROIs). We use LifeX as an annotation tool, which allows us to output nearly raw raster data (NRRD) files for image feature extraction. However, due to technical problems with LifeX, we had ROI annotation errors in 3 benign and 13 malignant patients. Therefore, these 16 patients’ data were excluded for the following process. The final images (for following training and test usage) have a size of either 512 × 512 or 768 × 768, depending on the raw data. Notably, the radiomics features are extracted only in ROIs in the reconstructed images, not directly from LDCT raw images. In these LDCT images, 73 patients were scanned using equipment from TOSHIBA, 7 patients were scanned using equipment from SIEMENS (Munich, Germany), 143 patients were scanned using equipment from GE MEDICAL SYSTEMS (Barrington, IL, USA), and 2 patients were scanned using equipment from Philips (Amsterdam, The Netherlands) for contrast imaging. Figure 1 illustrates PNs, where benign and malignant nodules appear to be strikingly similar, rendering precise diagnoses challenging. In this study, all nodule contours are manually delineated by an experienced radiological technician.

For ease of understanding, Figure 2 shows the flowchart of this study. The details of every block in the flowchart are described in the following paragraphs.

### 2.2. Isotropic Voxel Normalization and Image Reconstruction

To explore the reproducibility and model performance of PN malignancy diagnoses using the ML method, we normalized the PS and ST of all of the LDCT images using bicubic interpolation. This reconstruction aims to achieve consistent spatial resolution across all LDCT images, making them isotropic in three axes. Furthermore, to evaluate the influence of different voxel sizes and spatial resolutions on the model and features, we reconstructed all images to various voxel sizes, including images with side lengths of 0.5, 0.625, 0.75, 1, 1.25, 1.5, 1.75, and 2 mm. We utilized bicubic interpolation to achieve isotropic voxels.

### 2.3. Radiomics and Feature Selection

We employed radiomics to extract 2112 features from the PN regions of the reconstructed images. Radiomics encompass various quantitative image feature extraction methods, including first-order statistics, shape-based, and texture-based methods [11]. Not all extracted features contribute significantly to PN malignancy diagnosis; therefore, feature selection is necessary and crucial. Given that PN malignancy is often related to its size [10], we intentionally excluded features with size information. Before feature dimension reduction, 14 features related to shape and size were manually excluded (see Table A1 in Appendix A). Moreover, 123 features with identical feature values and no discriminability were also manually excluded. The remaining 1975 features underwent further feature dimension reduction; this process it outlined in the following paragraph.

The image analysis problem could induce a feature selection problem. In order to select significant features for classification in our model, we employed various methods. Firstly, to identify features with significant differences in mean and mode among benign and malignant PNs, we initially utilized the following two statistical methods: the independent *t*-test [29] and the Wilcoxon rank-sum [30] test. Features without significant differences were then excluded, with the goal of reducing the number of input features for the model. To perform an independent *t*-test, the data must be normally distributed and have equal variance. Meeting these criteria is crucial for using the independent *t*-test. Therefore, we first examine normality tests [31] and Levene tests [32] to assess the equality of variances and the type of distribution for both benign and malignant pulmonary nodule groups. If the data met the criteria of a normal distribution and equal variance, we proceeded to use an independent *t*-test for analysis, otherwise, we employed the Wilcoxon rank-sum test. Features between two classes with *p*-values less than 10^−20^ are considered to be significantly different, which means they have good and distinguishable features. Second, we employed the well-known LASSO [33] algorithm. The LASSO algorithm effectively reduces coefficients to zero for those features with less contribution to classification, thus achieving feature selection. Third, we applied t-distributed stochastic neighbor embedding (t-SNE) [34] for dimension reduction of selected feature combinations to visualize the distribution of patients in a two-dimensional space. t-SNE, belonging to manifold learning, achieves dimension reduction and meanwhile preserves the local structure of data distribution. This t-SNE can be found for data visualization and dimension reduction in recent publications. The reason that LASSO can be used in feature selection is explained in Appendix B.

### 2.4. Support Vector Machine (SVM) and Hyperparameter Optimization

SVM is a classic classifier [35] known for its strong performance in various classification tasks, particularly in dealing with small datasets. It exhibits a better generalization ability compared to DL models when dealing with small dataset [36]. In our study, we compared four combinations of feature selection: (1) all features without exclusion, (2) features with a *p*-value less than 10^−20^, (3) features selected by LASSO, and (4) features selected by the t-SNE algorithm. Before inputting all features into the SVM model, we performed a Min-Max Normalization on all features to ensure that different features had comparable numeric ranges. This is a basic feature-normalization process. In SVM, the choice of kernel function, which determines the decision boundary, is crucial. We employed the Gaussian Radial Basis Function (RBF) kernel function for our SVM model because most feature distributions have a normal distribution after feature selection. In ML models, hyperparameter optimization is highly correlated with final model performance. We used Gaussian Bayesian optimization [37] to tune the hyperparameters of the SVM model to achieve a better performance.

### 2.5. K-Fold Cross-Validation and Model Performance Evaluation

To measure the model performance, we manipulated data for training and tested it as follows: We randomly sampled 80% of the benign and malignant data as training data, with the remaining 20% serving as test data. This sampling and test process was repeated 4000 times. We then evaluated the model’s performance on different voxel sizes and feature selection methods using various metrics including Balanced Accuracy, Weighted Sensitivity, Weighted Precision, Weighted F-score, and Weighted AUC. Equations (1)–(5) represent the formulas for these metrics. Finally, we conducted a 10-fold cross-validation to assess the model performance and plot ROC curves.
(1)$$\mathrm{Balanced}\;\mathrm{Accuracy}=\frac{1}{k}\sum_{i=0}^{k}\frac{TP_{i}}{\left(TP_{i}+FN_{i}\right)}$$
(2)$$\mathrm{Weighted}\;\mathrm{Sensitivity}=\frac{\sum_{i=0}^{k}w_{i}\times TP_{i}}{\sum_{i=0}^{k}w_{i}*(TP_{i}+FN_{i})}$$
(3)$$\mathrm{Weighted}\;\mathrm{Precision}=\frac{{\sum_{i=0}^{k}w_{i}\times TP}_{i}}{\sum_{i=0}^{k}w_{i}*(TP_{i}+FP_{i})}$$
(4)$$\mathrm{Weighted}\;F1\;\mathrm{score}=\frac{\sum_{i=0}^{k}w_{i}\times (\frac{2\times TP_{i}}{2\times TP_{i}+FP_{i}+FN_{i}})}{\sum_{i=0}^{k}w_{i}}$$
(5)$$\mathrm{One}-\mathrm{vs}-\mathrm{All}\;\mathrm{AUC}=\frac{\sum_{i=0}^{k}w_{i}\times {\mathrm{AUC}}_{i}}{\sum_{i=0}^{k}w_{i}}$$

## 3. Results

In Figure 3, we demonstrate the distribution of *p*-values for features extracted at different voxel sizes reconstructed from the raw image data (LDCT). The value started from 0.5, and 0.625 to 2 is the side length of the isotropic voxel in mm. The word ‘original’ means there is no reconstruction, it is raw LDCT data. The value started from 200, and 400 to 1600 is the number of features. We categorized *p*-values into four groups: less than 0.05, 1 × 10^−10^, 1 × 10^−20^, and 1 × 10^−28^. Interestingly, Figure 3 shows that the majority of features have *p*-values in the range: [0.05, 1 × 10^−10^]. However, a *p*-value in this range is not capable of distinguishing or classifying. Therefore, for the most part, features are useless. We also found that features extracted from the original LDCT images without uniform voxel sizes had the least useful features. Here the useful features are indicated by the purple part of the circle in Figure 3 (*p*-value < 1 × 10^−28^). This indicates that voxel normalization indeed affects feature extraction. Through this voxel normalization process using image reconstruction, the extracted image features exhibit significant statistical differences between benign and malignant nodules. This result confirms our hypothesis. Notably, features extracted from reconstructed data normalized to a side length of 2 mm performed poorly compared to other side lengths. The exact quantity of features plotted in Figure 3 can be seen in Table 1. 

In Table 2, we illustrate the number of features before (the baseline) and after feature selection. In Figure 4, we demonstrate the model performance for comparison using 5 metrics illustrated in Equations (1)–(5). The results shows that the best feature selection method is LASSO, which used only 11 features. Surprisingly, the second-best performance was achieved by reducing the dimensionality of 11 LASSO-selected features to 2 ‘features’ (i.e., 2 directions in the feature space) using t-SNE, resulting in an outstanding performance. The feature distribution of these 2 directions are shown in Figure 5.

Figure 6 depicts the performance of features extracted from different voxel sizes’ reconstructions. Here, we used LASSO to select features. From the figure, it is clear that the model has better performances with voxel normalization with side lengths less than 2 mm. The best model was one with features extracted from LDCT images with a side length of 1.5 mm.

In Figure 7, we show ROCs using ten-fold cross-validations with radiomic features selected by LASSO from image reconstruction with a side-length of 1.5 mm. The model average AUC reached 0.982, indicating a stable diagnostic performance. This is clinically acceptable for PN diagnoses. In Figure 5 we demonstrate the distribution of 2D feature space, with two dimension directions. After dimensionality reduction, only 2 features were selected by t-SNE. From this distribution, it is clear that these 11 features can effectively classify benign and malignant nodules.

Finally, in Table 3 we compare different results with different voxel sizes. These results are the averaged values of various metrics in different voxel sizes of reconstruction repeated over 4000 times. We also compare our results to previous state-of-the-art models. Our best result outperforms most previous methods and is comparable to the best previous study [24]. Indeed, deep learning approaches need more time in training than the traditional method.

Table 4 presents the 11 radiomic features selected by the LASSO in this study. Notably, 7 of the 11 selected features are texture-related, demonstrating their important role in discriminating between benign and malignant PNs. Therefore, voxel normalization is essential and has a profound impact on reproducibility.

## 4. Discussion

Due to the diversity of patients in clinics and orders from physicians, radiographers use various parameter settings to acquire LDCT images. If these parameter settings are not consistent, it is difficult to compare them due to vacancy on baseline. Particularly, we are referring to texture features; different pixel-size on images can cause different results, such as features extracted from Gray Level Co-occurrence Matrix (GLCM), Gray Level Run Length Matrix (GLRLM), Gray Level Size Zone Matrix (GLSZM), and other texture features. Many previous studies have indicated that texture features are crucial for distinguishing the benign and malignant nature of PNs.

In this research, normalizing the voxel size from LDCT images to 1–1.5 mm yields good model performance. Typically, CT has a spatial resolution ranging from 0.5 mm to 0.625 mm in the *x*-*y* axis; our collected data fall in the range of 0.6 mm to 0.8 mm. The resolution on the z-axis depends on the temporal resolution. Most images on the z-axis resolution fall in the range of 1 mm to 5 mm in our in-house dataset. We do not have data in the z-axis measuring less than 1 mm. The image reconstruction must consider the spatial and temporal resolution of the raw data; otherwise, it can lead to a partial volume effect and subsequently impacts the model performance. Further, reconstructing images with a voxel larger than 1.5 mm of side length will significantly decrease spatial resolution, making it impossible to capture fine texture of PNs. Therefore, we recommend that future studies consider voxel reconstruction within the range of 1–1.5 mm, based on a prerequisite that data has higher spatial and temporal resolution than 1 mm in the *x*-*y*-*z* axis, i.e., ≤1 mm.

In 2023, Fischbach et al. [38] conducted a study on the diagnosis of nodules with respect to slice thickness (SK). In [38], they mention that as the SK decreases, the image noise decreases, and contrast increases. The authors tested SKs of 0.625 mm, 1.25 mm, 2.5 mm, 3.75 mm, 5 mm, 7.5 mm, and 10 mm. They found that a slice thickness of 0.625 mm yielded the best image quality. However, the best diagnostic quality was achieved with a slice thickness of 1.25 mm. Their finding aligns with our research findings, once again demonstrating that voxel normalization to an appropriate range (1.25–1.5 mm) can yield the best diagnostic benefits, while excessively small or large SK can lead to a decline in diagnostic performance.

One advantage of our study compared to previous studies is that all of our patients had pathological diagnoses to confirm the PN’s nature, i.e., benign or malignant. This advantage is not present in large open datasets such as LIDC-IDRI. Research limitations regarding this study are discussed below. This study mainly analyzed Asian ethnic groups, and most malignant lesions are adenocarcinoma spectrum. Therefore, the predictive ability for other pathological types of lung cancer may be lower. Particularly, this study mainly distinguishes lung adenocarcinoma spectrum lesions from other benign lesions. Therefore, we did not conduct any further analysis on the clinical management of these nodules.

In this study, PN contours are manually drawn by experts. However, if the PN regions could be automatically segmented on the LDCT images, together with the proposed system, this could create a potential for in-clinic utilization. This will contribute to the futural development of a comprehensive end-to-end diagnostic system. Many previous studies are devoted to pulmonary nodule detection such as competitions using the LUNA2016 dataset [39]. LUNA2016 is a subset of LIDC-IDRI [5]. We also performed a study [40] on nodule detection using the complete LIDC-IDRI dataset. In the future, we will combine these two systems to form one comprehensive system.

In our study, a significant limitation arises from the relatively small dataset obtained from a single center. The exclusive collection of patient imaging data with pathology verification posed challenges in the data acquisition process. Furthermore, the need for experienced personnel to manually delineate Regions of Interest (ROIs) for each case added complexity to dataset acquisition. To overcome this limitation, our future efforts will focus on the continuous collection of imaging data with pathology verification and uniform voxel sizes from various centers. Our aim is to establish a comprehensive database tailored to the Asian population, providing a substantial resource for subsequent researchers conducting studies on PNs recognition.

## 5. Conclusions

This study used a ML method combined with radiomic features extracted from reconstructed images with voxel normalization from LDCT. We mainly explored the impact of voxel normalization to predict performance between benign and malignant pulmonary nodules. Our study offers a recommendation: before using radiomics, the voxel normalization is important and crucial to texture-related studies. The reconstruction must consider the limitation on raw data temporal resolution. According to our finding, the best prediction of nodule classification, benign or malignant, is achieved using an isotropic voxel with a side length of 1.5 mm.

## Figures and Tables

**Figure 1 diagnostics-13-03690-f001:**
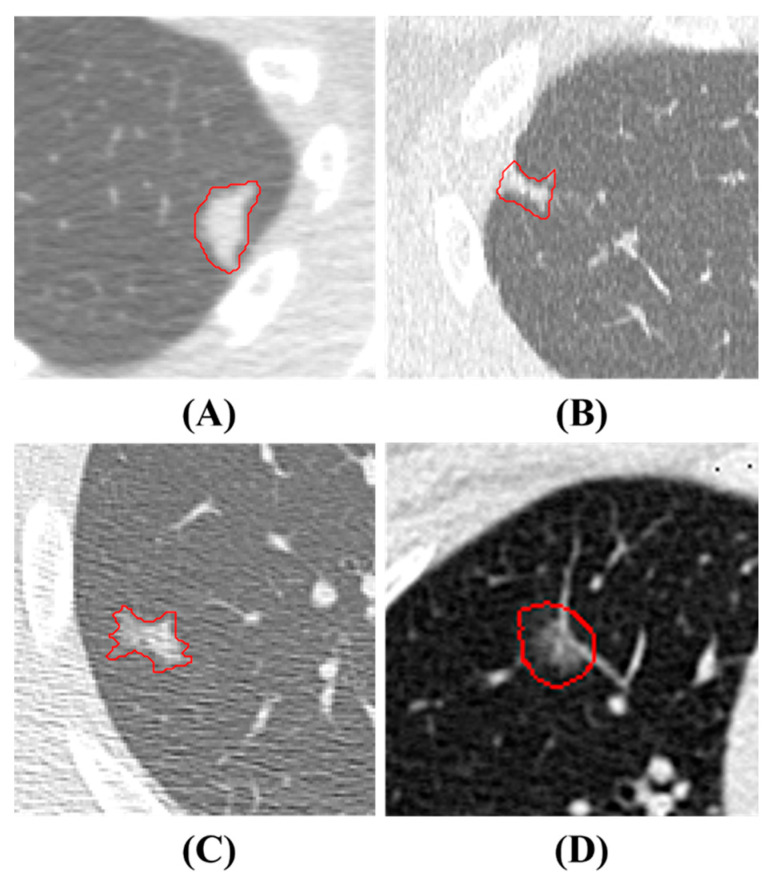
PNs Presentation. The red contours are manual delineations on nodules. (**A**) and (**B**) depict benign PNs, while (**C**) and (**D**) show malignant PNs.

**Figure 2 diagnostics-13-03690-f002:**
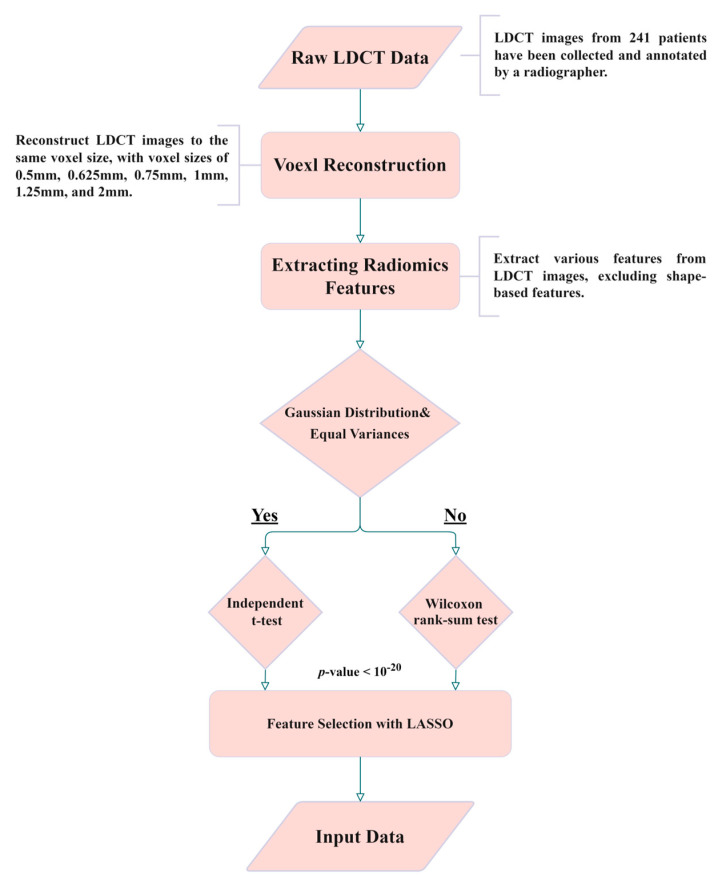
Experimental flowchart of this study.

**Figure 3 diagnostics-13-03690-f003:**
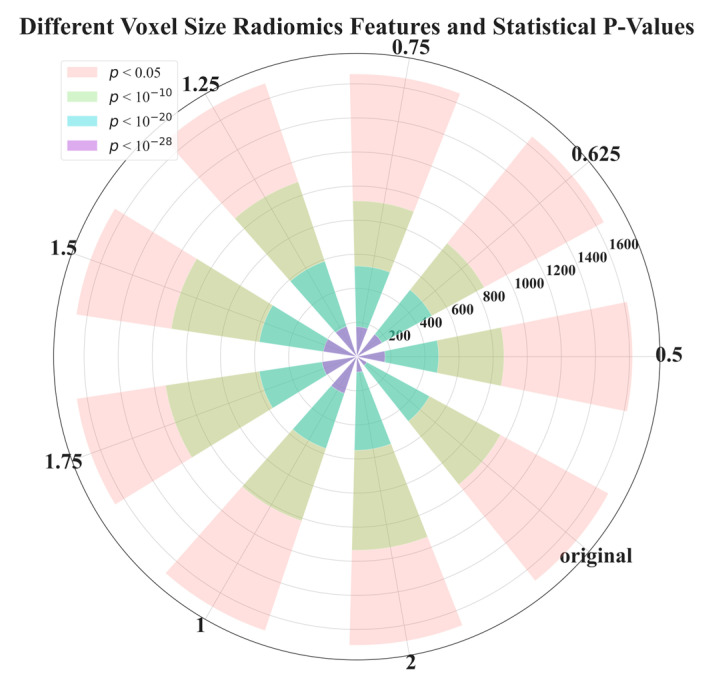
*p*-value distribution of radiomics features extracted from voxels with different side lengths reconstructed from raw LDCT data. This experiment aimed to examine the feature efficacy of benign and malignant nodule classification, based on different voxel sizes.

**Figure 4 diagnostics-13-03690-f004:**
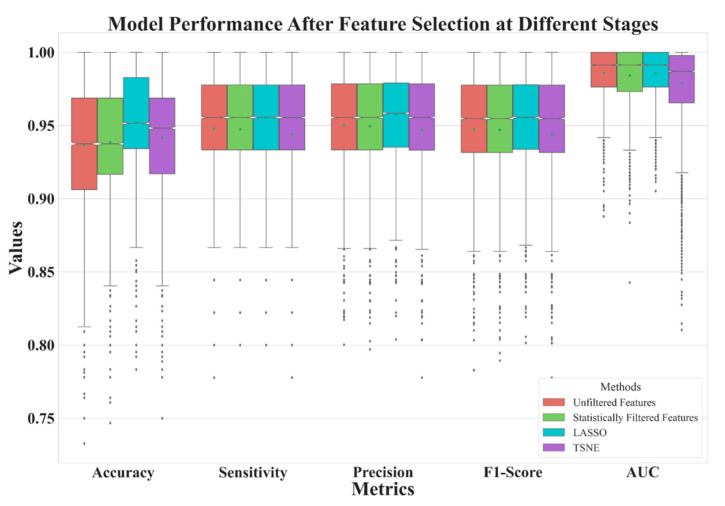
Model performance comparison of three feature selection methods and a baseline.

**Figure 5 diagnostics-13-03690-f005:**
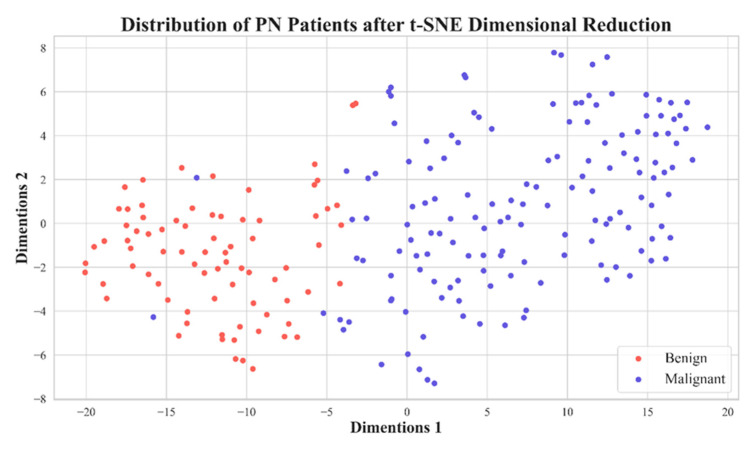
Distribution plot of 11 features extracted from PNs with a voxel with a side length of 1.5 mm after dimension reduction using t-SNE.

**Figure 6 diagnostics-13-03690-f006:**
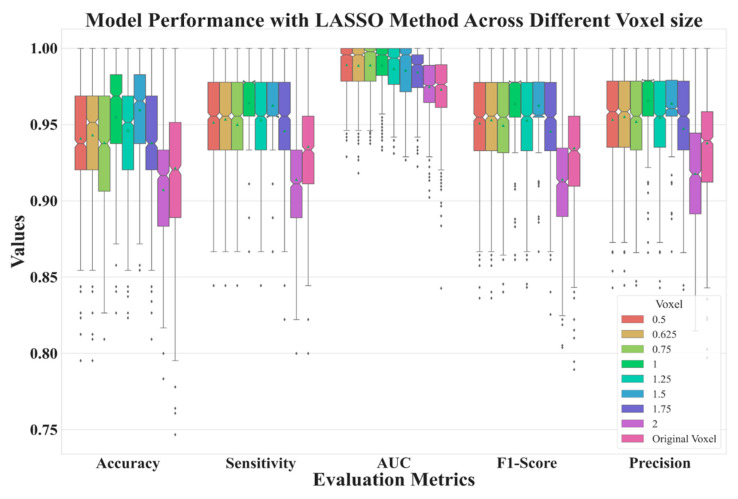
Model performance for reconstructed data with various side lengths.

**Figure 7 diagnostics-13-03690-f007:**
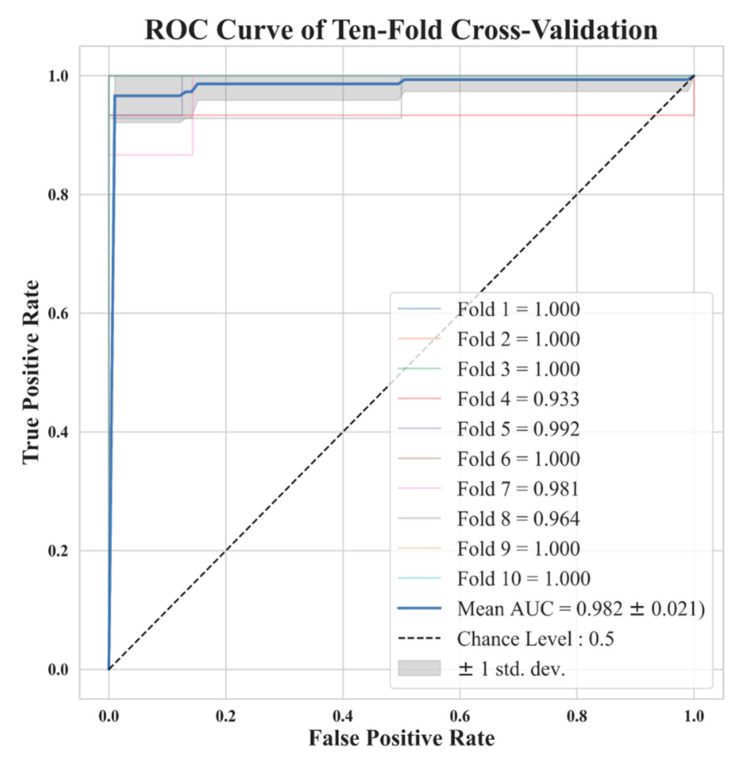
Ten-fold cross-validation ROC curve for the model trained with 11 features selected by LASSO with a voxel with a side length of 1.5 mm. Note that some lines are overlapping.

**Table 1 diagnostics-13-03690-t001:** The quantity of *p*-value of radiomics features distribution plotted in Figure 3.

Voxel Size	0.5	0.625	0.75	1	1.25	1.5	1.75	2	Original
*p* < 0.05	1617	1650	1657	1694	1690	1663	1661	1692	1680
*p* < 1 × 10^−10^	863	850	913	1016	1081	1100	1134	1135	959
*p* < 1 × 10^−20^	480	501	531	568	590	578	578	549	485
*p* < 1 × 10^−28^	166	168	175	227	187	198	206	91	67

**Table 2 diagnostics-13-03690-t002:** Number of features after feature selection and raw LDCT images.

	Unfiltered Features	Statistically Filtered Features	LASSO	t-SNE
Feature Number	2061	480	11	2

**Table 3 diagnostics-13-03690-t003:** Comparison of test results averaged from 4000 times; 80% training and 20% test in partition. The values in the first column denote different side lengths of voxels after image reconstruction.

	Accuracy	AUC	Sensitivity	Precision	F1 Score
0.5	0.9409	0.9891	0.9514	0.9533	0.9509
0.625	0.9431	0.9887	0.9535	0.9551	0.9530
0.75	0.9350	0.9890	0.9481	0.9501	0.9473
1	0.9531	0.9890	0.9624	0.9640	0.9620
1.25	0.9467	0.9866	0.9532	0.9548	0.9530
1.5	**0.9596**	**0.9855**	**0.9619**	0.9633	0.9619
1.75	0.9371	0.9844	0.9452	0.9468	0.9449
2	0.9073	0.9747	0.9156	0.9197	0.9159
Original	0.9223	0.9731	0.9357	0.9381	0.9349
Halder et al., 2021 [24]	**0.9610**	**0.9936**	**0.9685**	-	-
Mehta et al., 2021 [19]	-	0.8659	-	-	-
Shen et al., 2017 [22]	0.8612	-	-	-	-
Lu et al., 2021 [18]	0.934	0.984		-	-

**Table 4 diagnostics-13-03690-t004:** Description of the 11 Features Utilized by the Model.

Feature Description	Types
original_gldm_SmallDependenceLowGrayLevelEmphasis	Texture
log-sigma-2-0-mm-3D_glcm_DifferenceEntropy	Texture
log-sigma-2-0-mm-3D_gldm_SmallDependenceEmphasis	Texture
log-sigma-3-0-mm-3D_glszm_ZonePercentage	Texture
lbp-2D_gldm_DependenceNonUniformityNormalized	Texture
lbp-3D-m1_gldm_DependenceNonUniformityNormalized	Texture
lbp-3D-m2_gldm_DependenceNonUniformityNormalized	Texture
log-sigma-2-0-mm-3D_firstorder_Mean	First order
lbp-3D-m1_firstorder_Skewness lbp-3D-	First order
wavelet-LLH_firstorder_Mean	First order
wavelet-LHL_firstorder_Mean	First order

## Data Availability

Not applicable.

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
