# Peer review of "Impact of Voxel Normalization on a Machine Learning-Based Method: A Study on Pulmonary Nodule Malignancy Diagnosis Using Low-Dose Computed Tomography (LDCT)"

_diagnostics, 2023, doi:10.3390/diagnostics13243690_

Round 1

Reviewer 1 Report

Comments and Suggestions for Authors

The subject is interest. Many papers have been involved with the analysis of medical imaging. Except the interest of the paper, the athors must do a lot of work for publising their work

Suggestions:

1. Analysis of segmentation techniques and propose ways to segment the ROI (regions of interest) - use active contours or level sets methods

2. Compare different methods with their results. 

3. The flow chart must be analysed with more statistical formulations. For example the authors discuss the LASSO method. They must include more statistical methodology, why? how? comparison with original regression analysis.

4. In page 4 why the authors introduce 2 different methods for comparison parametric (t-test) and no parametric (Wilcoxon rank-sum)? Also the line <In order to reduce the number of input features for our model> it is not correct. These tests are used to find significan difference between mean or mode. If we want to reduce the number of inputs we propose factor analysis or princinal components or discriminal analysis.

In general the article needs a major work (some parts are not clear and some other are not correct)

Reviewer 2 Report

Comments and Suggestions for Authors

This paper explored the performance disparities of classification models trained on original images and LDCT images with different voxel sizes .  Using 11 features, a support vector machine (SVM) was trained to identify a nodule to be benign or malignant on LDCT images with an isotropic voxel having side-length of 1.5 mm for 225 patients in house. The result yields a favorable model performance with an accuracy of 0.9596 and an area under the receiver operating characteristic curve (ROC/AUC) of 0.9855. That sounds somewhat interesting, My main concerns are as follows:

1.Since the resolution in the Z-axis direction is different from that in the xy plane, how to obtain the isotropic voxel?

2. Different scanning layer thickness has a great impact on classification accuracy, and how different layer thickness affects classification accuracy can be analyzed and discussed.

3.The original image is interpolated to get a series of images of different voxel sizes, and then classified by support vector machine, will overfitting occur?

4. With multicenter scanners from different brands, the results should be more convincing.

Comments on the Quality of English Language

This paper explored the performance disparities of classification models trained on original images and LDCT images with different voxel sizes .  Using 11 features, a support vector machine (SVM) was trained to identify a nodule to be benign or malignant on LDCT images with an isotropic voxel having side-length of 1.5 mm for 225 patients in house. The result yields a favorable model performance with an accuracy of 0.9596 and an area under the receiver operating characteristic curve (ROC/AUC) of 0.9855. That sounds somewhat interesting, My main concerns are as follows:

1.Since the resolution in the Z-axis direction is different from that in the xy plane, how to obtain the isotropic voxel?

2. Different scanning layer thickness has a great impact on classification accuracy, and how different layer thickness affects classification accuracy can be analyzed and discussed.

3.The original image is interpolated to get a series of images of different voxel sizes, and then classified by support vector machine, will overfitting occur?

4. With multicenter scanners from different brands, the results should be more convincing.

5.The paper is rough, and the introduction and discussion need to be further improved.

Round 2

Reviewer 1 Report

Comments and Suggestions for Authors

The authors had made some changes but they are incomplete.

For example in Fig 1 the results from the 3 first images are based on a segmentation techniques based on which method? the D graph is not a result of segmentation. That is inconsistant.

That section 2 start with a graph. It is bettet the graph to be in another page. There is not presented the pseudo-code of the graph.

The authors said that they have implement the bicubic interpolation. Between which slices, thw distance between the slice was narrow, what are problema with this interpolation, graph of the interpolation process

The authors have written a appedix about LASSO but the question is how this estimaation process fit in their image analyis problem. Also the SVM needs more informations about the process, functions, graphical presentation

More analysis for the conclusion

Reviewer 2 Report

Comments and Suggestions for Authors

The authors have addressed my concerns and I recommend acceptance for publication.

The article needs a little tweaking:

1. Please move Figure 1 from the introduction to Materials and methods.

2. Please move Table 4 from the discussion section to the results section.
